# Metabolic Profiling of SH-SY5Y and Neuro2A Cells in Relation to Fetal Calf Serum (FCS) Concentration in Culture Media

**DOI:** 10.3390/metabo14040188

**Published:** 2024-03-26

**Authors:** Lys Kronenberger, Janine Mett, Jessica Hoppstädter, Uli Müller

**Affiliations:** 1Zoology/Physiology-Neurobiology, ZHMB (Center of Human and Molecular Biology), Faculty NT-Natural Science and Technology, Saarland University, D-66123 Saarbrücken, Germany; 2Department of Pharmacy, Pharmaceutical Biology, Saarland University, D-66123 Saarbrücken, Germany

**Keywords:** metabolism, physiological cell culture, glucose, glutamine, pyruvate, lactate, SH-SY5Y, Neuro2A

## Abstract

The neuroblastoma cell lines SH-SY5Y and Neuro2A are commonly utilized models in neurobiological research. DMEM supplemented with different nutrients and 5–10% Fetal Calf Serum (FCS) is typically used for culturing these cell lines. During special treatments, a reduced FCS content is often deployed to reduce cellular proliferation or the content of bioactive compounds. The impact of the reduction of FCS in culture media on the metabolic profile of SH-SY5Y and Neuro2A cells is currently unknown. Using an Amplex Red Assay, this study showed that the consumption of L-glutamine decreased after FCS reduction. Glucose and pyruvate consumption increased in both cell lines after the reduction of FCS. Thus, lactate production also increased with reduced FCS concentration. The reduction of FCS in the cell culture medium resulted in a reduced aerobic ATP production for SH-SY5Y cells and a complete shut down of aerobic ATP production for Neuro2A cells, measured using the Seahorse XF Real-Time ATP Rate Assay. Utilizing the Seahorse XF Glutamine Oxidation Stress Test, Neuro2A cells showed an increased utilization of L-glutamine oxidation after reduction of FCS. These results indicate that changes in FCS concentration in culture media have an impact on the different energy production strategies of SH-SY5Y and Neuro2A cells which must be considered when planning special treatments.

## 1. Introduction

Mammalian cell cultures play a pivotal role in advancing our understanding of the cellular, molecular, and biochemical properties of living cells. Cultivating mammalian cell lines involves using different media, and the choice significantly impacts cell growth and differentiation. The media usually contain excess energy substrates such as glucose, pyruvate, and amino acids to prevent cell starvation. This facilitates cell culture practice but creates non-physiological culture conditions, potentially affecting cellular metabolism and experimental outcomes [1].

Glycolysis and oxidative phosphorylation (OXPHOS) are two major metabolic pathways that provide cells with energy. After entering the cell, glucose is phosphorylated to glucose-6-phosphate (G6P), which is subsequently routed in the glycolytic pathway and pentose phosphate pathway (PPP). Glycolysis, which takes place in the cytosol, converts glucose into pyruvate, resulting in the reduction of nicotine amide adenine dinucleotides (NAD+) into NADH and the production of 2 molecules adenosine triphosphate (ATP) per molecule glucose. The generated pyruvate is then either actively transported into the mitochondria or turned into lactate and released from the cell. Within the mitochondria, pyruvate undergoes metabolization through the tricarboxylic acid (TCA) cycle and OXPHOS in the electron transport chain (ETC), yielding carbon dioxide (CO_2_) while consuming oxygen (O_2_) and approximately 30 molecules ATP per molecule of glucose, a significantly higher yield compared to the glycolytic pathway alone [2,3,4]. Besides glucose and pyruvate, amino acids, like L-glutamine, have been identified as essential nutrients for the in vitro cultivation of cells. In some cases, L-glutamine compensates for glucose to maintain TCA cycle function. L-glutamine enters the cell and is converted to L-glutamate by mitochondrial Glutaminase 1 (GLS1). A two-step reaction converts L-glutamate to alpha-Ketoglutarate, which is then introduced into the TCA cycle, where oxalacetate and further citrate can be formed [5]. Depending on the cell type, the utilization of these energy substrates available in the cell culture medium can vary.

The cell lines SH-SY5Y and Neuro2A are widely used models in neurobiological research. SH-SY5Y cells, originating from human neuroblastoma, provide insights into human properties, particularly in the context of neurodegenerative diseases and drug development. Neuro2A, murine neuroblastoma cells, display rapid proliferation and neurotypical differentiation, rendering them valuable for exploring neural differentiation pathways and neurodevelopmental processes [6,7,8,9,10]. Despite their common use in research, relatively little is known about the energy metabolism of these cell lines, a crucial aspect closely linked to cellular physiology. The availability of energy substrates impacts the formation of essential components like NADPH, ATP, or the generation of new biomass and thus on the proliferation [11]. Furthermore, the energy status of the cells influences signaling pathways such as that of the AMPK, which is related to many metabolic disorders such as Diabetes mellitus or Morbus Alzheimer and is therefore of general interest in research [12,13].

The current literature shows how different cell culture media with various compositions affect the consumption of the available energy substrates and energy production strategies of the cell lines investigated here. Generally, only one energy substrate was in focus [14,15,16]. The effects of the FCS concentration in the cell culture medium were not considered concerning the impact on cellular metabolism, even though FCS contains important energy substrates, like glucose or amino acids [17].

In this study, our objective was to comprehensively characterize the energetic phenotype under different FCS concentrations in cell culture media exhibited by SH-SY5Y and Neuro2A cells. To achieve this, we systematically quantified the rates of glucose and pyruvate uptake, lactate release, the demand for L-glutamine, and the ATP production arising from glycolysis and mitochondrial respiration for both cell lines under different physiological culture conditions.

## 2. Materials and Methods

All chemicals and cell culture materials were purchased from Merck/Sigma-Aldrich (Darmstadt, Germany) if not stated otherwise. All chemicals, devices, and software which were utilized for the performance and analysis of the Seahorse XF Assays were purchased from Agilent (Waldbronn, Germany) if not stated otherwise.

### 2.1. Cell Culture

Human SH-SY5Y cells were purchased from Merck/Sigma-Aldrich (Darmstadt, Germany). Murine Neuro2A neuroblastoma cells were obtained from CLS Cell Lines Service (Eppelheim, Germany). Cells were maintained in phenol red-free Dulbecco’s Modified Eagle Medium (DMEM) containing 5% fetal calf serum (FCS), 8.25 mM glucose, 1% non-essential amino acid solution, 4 mM L-glutamine, 2 mM sodium pyruvate, penicillin (100 U/mL)/streptomycin (0.1 mg/mL), and 0.05 mg/mL gentamicin (DMEM/5% FCS) in a humidified atmosphere containing 5% CO_2_ at 37 °C. Cells were passaged before reaching confluency by detachment with trypsin/EDTA. For experiments, cells were seeded onto 96-well cell culture plates as described below.

### 2.2. Quantification of Glucose, Pyruvate, L-Glutamine, and Lactate in Culture Media Using Amplex Red Assays

Cells were seeded onto transparent 96-well cell culture plates and incubated in DMEM/5% FCS for 24 h. Based on the effects of FCS on cell proliferation [18,19], the initial cell numbers were as follows: 1.5 × 10^4^ Neuro2A and 2 × 10^4^ SH-SY5Y cells per well for the DMEM/5% FCS condition, and 2 × 10^4^ Neuro2A and 3 × 10^4^ SH-SY5Y cells per well for the DMEM/0.1% FCS condition, to achieve equal cell numbers at the start of measurements. The next day, the medium was replaced by 100 µL fresh culture medium, comprising either 5% FCS (DMEM/5% FCS) or 0.1% FCS (DMEM/0.1% FCS). Media samples (4 µL) were collected from each well immediately (0 h) and after 12, 24, and 48 h. Samples were diluted in 100 mM potassium phosphate pH 7.4 to achieve a final 1:500 dilution for glucose, pyruvate, and lactate measurements. For quantification of L-glutamine, media samples were diluted 1:25 in 25 mM sodium acetate buffer (pH 4.7). Media samples were stored at −20 °C before usage.

To determine cellular nutrient uptake and lactate production rates, the concentrations of glucose, pyruvate, L-glutamine, and lactate in the media samples were quantified. The measurements were conducted using various Amplex Red (10-Acetyl-3,7-dihydroxyphenoxazine) assays. These assays rely on the oxidation of glucose, pyruvate, or lactate by specific oxidases, with subsequent quantification of hydrogen peroxide (H_2_O_2_) as a by-product. For quantification of L-glutamine, L-glutamine was converted into L-glutamate by glutaminase, followed by quantification of L-glutamate using glutamate oxidase. In the presence of horseradish peroxidase (HRP), H_2_O_2_ reacts with Amplex Red in a 1:1 stoichiometry, producing the fluorescent oxidation product resorufin. Resorufin has fluorescence excitation and emission maxima of approximately 571 nm and 585 nm [20,21].

For measurements, samples were thawed, transferred onto black 96-well plates, and utilized in Amplex Red assays as follows. For the Amplex Red glucose and lactate assay, 25 µL of the samples and 50 µL of potassium phosphate pH 7.4 were pipetted onto a black 96-well plate. In the case of pyruvate measurement, 12.5 µL of the samples and 62.5 µL of potassium phosphate pH 7.4 were used. For the L-glutamine assay, 20 µL of the samples were transferred onto black 96-well plates and incubated for 2 h at 37 °C with 5 µL of glutaminase. As a reference, standards with known glucose/pyruvate/L-glutamine/lactate concentrations were equally diluted and simultaneously measured. Immediately before starting the measurements, 25 µL of a reaction mix containing 100 µM Amplex Red (Biomol, Hamburg, Germany), 0.2 U/mL HRP (ThermoFisher Scientific, Kandel, Germany), together with either 2 U/mL glucose oxidase, 1 U/mL pyruvate oxidase, or 0.005 U/mL glutamate oxidase or 0.125 U/mL lactate oxidase were added. At 120-s intervals, the fluorescence signal of resorufin was determined at an excitation wavelength of 530 ± 17 nm and an emission wavelength of 590 ± 17 nm under light exclusion in a Safire^2^ Fluorometer (Tecan, Crailsheim, Germany). The increase in fluorescence over time (20 min for glucose and L-glutamine, and 10 min for pyruvate and lactate) was calculated for each well and utilized for further data analysis. The absolute glucose, pyruvate, L-glutamine, and lactate concentrations were calculated based on the linear regression of slope values of the corresponding standard curve.

### 2.3. Assessment of Cellular Energy Metabolism Utilizing the Seahorse XF Real-Time ATP Rate Assay

The Seahorse XF Real-Time ATP Rate Assay Kit was employed to measure the oxygen consumption rate (OCR) and the extracellular acidification rate (ECAR), pivotal indicators of mitochondrial respiration and glycolysis in live cells. Utilizing various respiration inhibitors, the assay enables the determination of total aerobic and anaerobic ATP production, according to the instructions of the Agilent Seahorse XF Real-Time ATP Rate Assay User Guide.

For the assay, cells (0.8 × 10^4^ Neuro2A and 2 × 10^4^ SH-SY5Y cells per well for the DMEM/5% FCS condition, and 1.2 × 10^4^ Neuro2A and 3 × 10^4^ SH-SY5Y cells per well for the DMEM/0.1% FCS condition) were seeded onto XF96 microplates and incubated in DMEM/5% FCS for 24 h. After removing the cell culture medium, 100 µL fresh culture medium containing either 5% FCS (DMEM/5% FCS) or 0.1% FCS (DMEM/0.1% FCS) was added for an additional 24 h. Then Seahorse XF Real-Time ATP Rate Assays were performed according to the manufacturer’s instructions (Standard Assay procedure). 1 h before measurement using an Agilent Seahorse XF Pro analyzer, the culture medium in all wells was replaced with XF DMEM Assay Medium (8.25 mM XF Glucose, 2 mM XF Pyruvate, 4 mM XF L-Glutamine, pH 7.4) and moved into a CO_2_-free incubator.

### 2.4. Determination of the Cellular L-Glutamine Demand Utilizing the Seahorse XF Substrate Oxidation Stress Test

The cellular oxidation of L-glutamine was measured based on changes in the oxygen consumption rate (OCR) using the Seahorse XF Substrate Oxidation Stress Test Kit (Agilent, Waldbronn, Germany). Using specific inhibitors of the L-glutamine pathway and the mitochondrial respiration, the cellular L-glutamine demand could be assessed by the basal and maximal respiration rate. The assay was performed according to the Agilent XF Substrate Oxidation Stress Test Kit User Manual.

Cells were seeded and treated as described above for the Seahorse XF Real-Time ATP Rate Assay. The Seahorse XF Substrate Oxidation Stress Test was then performed as described in the manufacturer’s instructions (Standard Assay procedure). DMEM/5% FCS or DMEM/0.1% FCS was replaced 1 h before the measurement using Agilent Seahorse XF Pro analyzer, with XF DMEM Assay Medium (8.25 mM XF Glucose, 2 mM XF Pyruvate, 4 mM XF L-Glutamine, pH 7.4) and moved into a CO_2_-free incubator. Bis-2-(5-phenylacetamido-1,3,4-thiadiazol-2-yl)ethylsulfid (BPTES) in a final concentration of 3 µM was used for the specific inhibition of Glutaminase 1 (GLS1), which converts L-glutamate from L-glutamine. Carbonyl-Cyanide-4-(trifluoromethoxy)-phenylhydrazone (FCCP), a mitochondrial uncoupler which allows the measurement of maximal OCR, was used in a final concentration of 2 µM.

### 2.5. In-Situ Cell Counting Using a BioTek Cytation 5

To visualize, image, and quantify the cells, a Cytation 5 system (BioTek Instruments, Bad Friedrichshall, Germany), which integrates automated microscopy with conventional microplate detection, was employed.

### 2.6. Calculation of Glucose Utilization, Pyruvate, and L-Glutamine Consumption, and Lactate Production Rates

To determine nutrient consumption and lactate production, the quantity of glucose/pyruvate/L-glutamine consumed, and the amount of lactate released by the cells within the initial 24 h, were divided by the incubation time of 24 h and the average cell count per well at 24 h.

### 2.7. Calculation of Total Aerobic and Anaerobic ATP Rate

Using Wave software, the OCR in pmol/min, ECAR in mpH/min, and Proton Efflux Rate (PER) in pmol/min, based on the ECAR, were determined. These parameters were normalized on the cell number per well quantified by the Cytation 5 system (BioTek Instruments, Bad Friedrichshall, Germany). Total aerobic (MitoATP) and anaerobic ATP production level (GlycoATP), using Microsoft Excel version 16.78.3 software, were calculated based on the kinetic curves of OCR and ECAR (Appendix A), according to the equations in the Seahorse XF Real-Time ATP Rate Assay Kit User Guide.

### 2.8. Calculation of the Cellular L-Glutamine Demand

Wave software was used to determine the OCR in pmol/min per well. The OCR was normalized on the cell number per well, which was quantified by the Cytation 5 system (BioTek Instruments, Bad Friedrichshall, Germany). The calculation of the respiration rates of the L-glutamine oxidation, utilizing Microsoft Excel version 16.78.3 software, was based on data interpretation by [22]. For the basal respiration rate, the third baseline measurement of the kinetic curve of OCR was used, as it was considered the most stable. The maximal respiration rate was calculated utilizing the first FCCP point, as it was the highest response to the inhibitor. The third measurement point after Rotenone/Antimycin A injection, defined as non-mitochondrial oxidation, was subtracted from the basal and maximal respiration rates. The acute response was defined as the sixth point after BPTES injection, deemed as the highest response, subtracted by the third baseline point (Appendix A).

### 2.9. Statistical Analysis

Statistical analyses were performed using OriginLab 2023b software (OriginLab Corporation). The results are presented as mean ± standard deviation (StDev). The number of wells per experimental group (n) is indicated in the legend. Values that were out of a range of two standard deviations were eliminated from the data. Determination of statistical significance consisted of unpaired Student´s *t*-Test with independent samples between media supernatant samples of glucose, pyruvate, L-glutamine, and lactate and was performed for Seahorse XF Assays measuring ATP rate or L-glutamine oxidation between DMEM/5% FCS and DMEM/0.1% FCS. Significance levels for p-levels were marked as follows: * ≤ 0.05, ** ≤ 0.01, and *** ≤ 0.001.

## 3. Results

### 3.1. Energy Substrate Consumption of Neuro2A and SH-SY5Y Cells Depends on the FCS Concentration in the Cell Culture Medium

To characterize the energy metabolism of SH-SY5Y and Neuro2A cells under different culture conditions, cells were incubated for up to 48 h starting with DMEM/5% FCS or DMEM/0.1% FCS, at physiological glucose concentration (8.25 mM) [23]. No media change was performed during the incubation period.

The concentration of the energy substrates glucose, pyruvate, L-glutamine, and lactate in cell culture media supernatant samples was determined at 0, 12, 24 and 48 h. Based on these data, the time points at which a biochemical hypoglycemia (<4 mM) or a lethal lactate concentration for humans (>5 mM) occurs could be identified [23,24].

For methodical control, concentrations of the energy substrates in the media of cell-free wells were measured. There were no differences in the concentrations over the time (Appendix A). This allowed all measured changes in the energy substrate concentrations of the media to be traced back to consumption by the cells.

Our results revealed that the concentration of glucose in media of human SH-SY5Y decreased continuously with a significantly lower glucose concentration after 24 h of SH-SY5Y cells treated with DMEM/0.1% FCS than treated with DMEM/5% FCS (*p* = 0.0376). After 48 h, the glucose concentration in the media of SH-SY5Y cells achieved a critical concentration <4 mM with both FCS concentrations. Lactate increased continuously and achieved a critical concentration of ≥5 mM after only 24 h. At this time point, SH-SY5Y cells treated with DMEM/0.1% FCS produced significantly more lactate than cells treated with DMEM/5% FCS (*p* = 0.0071) (Figure 1a). SH-SY5Y cells treated with DMEM/0.1% FCS consumed no L-glutamine, whereas cells treated with DMEM/5% FCS showed significantly higher consumption of L-glutamine at 12 h, 24 h, and 48 h (DMEM/5% FCS: 2.58 ± 0.45 mM; DMEM/0.1% FCS 3.06 ± 0.51 mM) (12 h: *p* < 0.001; 24 h: *p* < 0.001; 48 h: *p* < 0.001). While pyruvate was not consumed by SH-SY5Y cells treated with DMEM/0.1% FCS, cells treated with DMEM/5% FCS released pyruvate into the medium, resulting in a significantly higher measurable pyruvate concentration starting at 12 h (*p* = 0.0054) (Figure 1b).

Glucose concentration in the media of the murine cell line Neuro2A decreased sharply and reached concentrations below 4 mM after 24 h with both FCS concentrations. In Neuro2A cells, lactate increased rapidly and achieved the critical concentration of ≥5 mM after only 12 h. After 48 h, Neuro2A cells treated with DMEM/0.1% FCS showed a significant higher production of lactate compared to cells treated with DMEM/5% FCS (*p* < 0.001) (Figure 1c). The L-glutamine concentration in the media of Neuro2A cells decreased continuously. After 12 h, cells treated with DMEM/5% FCS consumed significantly more L-glutamine than cells treated with DMEM/0.1% FCS (*p* = 0.0104). L-glutamine concentrations smaller than 1 mM were achieved after 48 h for DMEM/5% FCS. The pyruvate concentration decreased continuously with both FCS concentrations. Neuro2A cells treated with DMEM/0.1% FCS reached a significantly lower pyruvate concentration after 24 h compared to cells treated with DMEM/5% FCS (*p* < 0.001). The critical pyruvate concentration (<1 mM) was reached in the media of cells treated with DMEM/5% FCS after 48 h. These findings indicate that Neuro2A cells treated with DMEM/0.1% FCS use more pyruvate than L-glutamine, whereas cells with DMEM/5% FCS use both energy substrates in parallel (Figure 1d).

For the calculation of the energy substrate consumption rates for the time window of 0–24 h, the measured concentrations in mM at 24 h were converted to fmol and then divided by the number of cells measured after 24 h. The time point of 24 h was chosen because most of the energy substrates were consumed within this time window.

Calculations revealed that SH-SY5Y cells treated with DMEM/0.1% FCS showed a lower glucose consumption rate than those treated with DMEM/5% FCS. The equal could be observed for the L-glutamine consumption rate. The lactate production rate of SH-SY5Y cells with DMEM/0.1% FCS was around 20% higher than that of cells treated with DMEM/5% FCS. Cells treated with DMEM/5% FCS also produced 250% more pyruvate compared to DMEM/0.1% FCS.

The glucose consumption rate of Neuro2A cells was around 10% higher with DMEM/0.1% FCS than with DMEM/5% FCS. This was also reflected in a 14% higher lactate production rate with DMEM/0.1% FCS. The higher consumption rate of cells treated with DMEM/0.1% FCS could also be observed for pyruvate, with 16% more pyruvate consumption by cells treated with DMEM/0.1% FCS. Neuro2A cells treated with DMEM/5% FCS consumed 33% more L-glutamine per cell per hour (Table 1).

These data clearly showed that SH-SY5Y cells reduced their energy substrate consumption rates after FCS reduction but increased their lactate production. In contrast, Neuro2A cells showed a boosted energy substrate consumption and an elevated lactate production if grown under reduced FCS concentration.

### 3.2. FCS Reduction Leads to a Switch in the Energy Production Strategy of Neuroblastoma Cell Lines

Glucose, pyruvate, L-glutamine, and lactate, as energy substrates, are metabolized by glycolysis or mitochondrial oxidative phosphorylation, the two main metabolic pathways to produce ATP. A Seahorse XF Real-Time ATP Rate Assay was performed to identify the contribution of the main metabolic pathways for ATP production after 24 h. Before measurements, cells received fresh assay medium with 8.25 mM glucose, 4 mM L-glutamine, and 2 mM pyruvate so that the energy substrates were completely available again.

In SH-SY5Y cells, a pretreatment with reduced FCS concentration resulted in a significant decrease of total ATP of 27% with DMEM/0.1% FCS (DMEM/5% FCS: 709 ± 104 pmol ATP/min; DMEM/0.1% FCS: 522 ± 142 pmol ATP/min) (*p* < 0.0001). The anaerobic ATP production decreases by 40% with DMEM/0.1% FCS compared to cells treated with DMEM/5% FCS (*p* < 0.001). The mitochondrial-produced ATP remained unchanged (Figure 2a). Thus, FCS reduction for 24 h resulted in an SH-SY5Y cell phenotype characterized by a more quiescent energy profile (Figure 2c).

FCS reduction in the media of Neuro2A cells led to a significantly decreased aerobic ATP production from 14% to 2% (*p* < 0.001). In contrast, anaerobic ATP production was significantly increased by 14% for DMEM/0.1% FCS compared to DMEM/5% FCS (*p* < 0.001). Total ATP production was not altered in this cell line (DMEM/5% FCS: 487 ± 61 pmol ATP/min; DMEM/0.1% FCS: 495 ± 77 pmol ATP/min) (Figure 2b). FCS reduction therefore changed the energetic phenotype of the Neuro2A cells to a more quiescent and glycolytic phenotype (Figure 2c).

These results indicate that the two neuroblastoma cell lines, SH-SY5Y and Neuro2A, utilize glycolysis as the main metabolic pathway to produce their ATP. FCS reduction in the cell culture medium leads to a switch in the energy production strategy, which varies depending on the cell line. FCS reduction in SH-SY5Y cells decreases the production of glycolytic ATP, while mitochondrial ATP production remains unchanged. FCS reduction in Neuro2A cells diminishes mitochondrial ATP production to zero, a reduction that is compensated by an enhanced glycolytic ATP production.

### 3.3. Neuro2A Cells Prefer L-Glutamine as an Alternative Energy Substrate after Reducing FCS in the Cell Culture Medium

Neurons and neuronal cell lines can use alternative energy substrates for mitochondrial energy production, for example amino acids. In nutrient-deficient environments, neurons switch to glutamine oxidation as an alternative to glucose [25]. Thereby, glutamine supplies carbon sources for energy production or can be used as nitrogen and carbon sources to synthesize essential metabolites [26].

Quantification of the L-glutamine concentration using an Amplex Red Assay showed that both neuroblastoma cell lines consumed L-glutamine. To now determine the capacity of the neuroblastoma cell lines to utilize glutamine oxidation as an alternative energy production source during the fasting state, a Seahorse Glutamine Oxidation Stress Test was performed after 24 h of incubation with DMEM/5% FCS or DMEM/0.1% FCS. Before measurements, cells received fresh assay medium so that the energy substrates were completely available again.

The basal respiration before the inhibition of L-glutamine oxidation showed a mitochondrial respiration rate of about 40 pmol/min for SH-SY5Y cells and a mitochondrial respiration rate averaging 20 pmol/min for Neuro2A cells. Reduction of FCS in media had no effects on the basal mitochondrial respiration (Figure 3). The tendency for higher basal mitochondrial respiration (OCR) of SH-SY5Y cells compared to Neuro2A cells is consistent with the results of the measured mitochondrial ATP production levels and the energetic profiling (Figure 2).

The acute mitochondrial response of the neuroblastoma cells to Glutaminase 1 (GLS1) inhibitor BPTES, which inhibits the oxidation of L-glutamine leading to a L-glutamate deficiency, led to a decrease of the mitochondrial respiration rate, averaging −1.5 pmol/min for SH-SY5Y cells (Figure 3a). The mitochondrial respiration rate of Neuro2A cells measured as about −4 pmol/min. FCS reduction in the culture media had no effects on this (Figure 3b).

To measure the maximal possible activity of the mitochondrial respiration chain to metabolize L-glutamine (maximal respiration), the mitochondrial uncoupler FCCP was used. The mitochondrial substrate oxidation is thereby decoupled from the ATP synthesis, which removes the rate limit imposed by basic cellular demand for ATP [27]. Maximal respiration of SH-SY5Y cells stayed the same compared to the basal level for DMEM/5% FCS, and increased minimally from 36 pmol/min to 45 pmol/min with DMEM/0.1% FCS (Figure 3a). Maximal respiration of Neruo2A cells increased from a basal respiration of 22 pmol/min to 60 pmol/min for DMEM/5% FCS, and from 23 pmol/min to 80 pmol/min for DMEM/0.1% FCS. A significant increase in L-glutamine utilization can be concluded after the reduction of FCS in the culture media of Neruo2A cells (*p* = 0.0093) (Figure 3b).

These findings indicate that human SH-SY5Y show a minor dependence on L-glutamine oxidation, whereby a reduction of FCS had no effects. Murine Neuro2A cells had a high preference to use L-glutamine oxidation as an alternative pathway for mitochondrial energy production because they showed a high sensitivity to GLS1 inhibition by BPTES and high maximal respiration after injection of the mitochondrial uncoupler FCCP. The dependence on L-glutamine oxidation of Neuro2A cells was amplified with a reduction of FCS.

## 4. Discussion

The present study showed that the neuroblastoma cells SH-SY5Y and Neuro2A, which are widely used models in neurological research, differ with respect to their general energy production strategies and their adaption to changes in FCS concentration in the cell culture medium.

Utilizing DMEM containing 5% FCS, human SH-SY5Y cells produce two-third of their ATP by glycolysis (GlycoATP), with one-third through mitochondrial OXPHOS. The high glucose consumption together with the sharply increasing lactate production point to a Warburg Effect. In the presence of 5% FCS, SH-SY5Y cells consumed L-glutamine and released pyruvate into the medium. Reduction of FCS to 0.1% changed the energy phenotype of SH-SY5Y cells to a more quiescent one, accompanied by a 40% reduction of GlycoATP production. If grown in DMEM/0.1% FCS, SH-SY5Y cells consumed glucose more rapidly, associated with a faster increase in lactate production. This indicates that an FCS reduction strengthened the Warburg Effect. Consumption of L-glutamine and the release of pyruvate into the medium is reduced when compared to cells grown in DMEM/5% FCS. This indicates that SH-SY5Y cells grown in DMEM/5% FCS can extract certain energy substrates from FCS, which can be converted to pyruvate, which is then released by the cells. As a potential scenario, amino acids could be inserted into the TCA cycle leading to the generation of malate, which can be oxidized into pyruvate and released [28]. This additional source of energy substrates is limited by reducing the FCS to 0.1% so that less pyruvate is released by the cells. FCS contains energy substrates like glucose, amino acids, and fatty acids [17].

The fact that the composition of the cell culture medium has an influence on the metabolism of the cells also becomes clear when comparing the values measured in this study with those from the literature. If DMEM with a similar FCS concentration (10%) is used, the L-glutamine consumption of the SH-SY5Y cells does not differ [14]. Using a different physiological medium such as Plasmax with a different composition of energy substrates for the cultivation of SH-SY5Y cells seems to increase their glucose consumption. Plasmax is a cell culture medium whose concentrations of more than 50 different nutrients and metabolites are adapted to the concentrations in human blood plasma [15,29]. Differentiation of SH-SY5Y cells by retinoic acid (RA), which is utilized to investigate a variety of diseases, seems to lead to an increased mitochondrial oxygen consumption rate and glycolysis compared to undifferentiated cells like those used in this study [16]. Based on the findings from the literature and this study, effects of certain treatments and media on the cellular metabolism should be factored in.

Murine Neuro2A cells produced ATP almost completely by glycolysis (77%) with minor amounts of MitoATP when grown in DMEM/5% FCS. Neuro2A cells mainly consumed glucose with a strongly increasing lactate production. This indicates that Neuro2A cells exhibit the Warburg Effect. Additionally, Neuro2A cells consumed L-glutamine and pyruvate and showed an increased rate of L-glutamine oxidation. If grown in medium with reduced FCS (0.1%), Neuro2A cells generate ATP only by glycolysis (92%), showing an increased glucose consumption and lactate production rate. This indicates that the FCS reduction enhances the Warburg Effect. Besides that, Neuro2A cells showed a reduced L-glutamine consumption, but Neuro2A cells after FCS reduction were able to oxidize more L-glutamine via mitochondrial respiration. Pyruvate consumption stayed the same after FCS reduction.

The enhanced Warburg Effect after FCS reduction suggests that 0.1% FCS leads to a limitation of energy substrates and that Neuro2A cells with 5% FCS resort to energy substrates from FCS. As a fast-growing cell line, Neuro2A cells are probably dependent on an additional energy pathway like fatty acid synthesis to ensure the synthesis of membranes and signaling molecules [19,30,31]. For fatty acid synthesis, cells need higher proportions of pyruvate than L-glutamine [26], which corresponds to the findings of this study.

To our knowledge there are no studies in the current literature to which our findings about the Neuro2A cell line can be compared.

## 5. Conclusions

In conclusion, reduction of FCS in cell culture media leads to changes in the energy production strategy and energy substrate consumption of neuroblastoma cell lines. The strategies to deal with the reduced FCS concentration in the medium vary depending on the cell line. SH-SY5Y cells reduce glycolytic ATP production with unchanged MitoATP, whereas Neuro2A cells switch completely to glycolytic ATP production. To maintain the cell culture conditions and thus the metabolism of the cells, the novel findings of this study should be factored in during cultivation and when choosing a cell line for a neuronal research project. SH-SY5Y cells require a medium grown in DMEM/5% FCS after 48 h and a medium change after FCS reduction is required 24 h earlier. Neuro2A cells always require fresh medium after 24 h regardless of FCS reduction.

## Figures and Tables

**Figure 1 metabolites-14-00188-f001:**
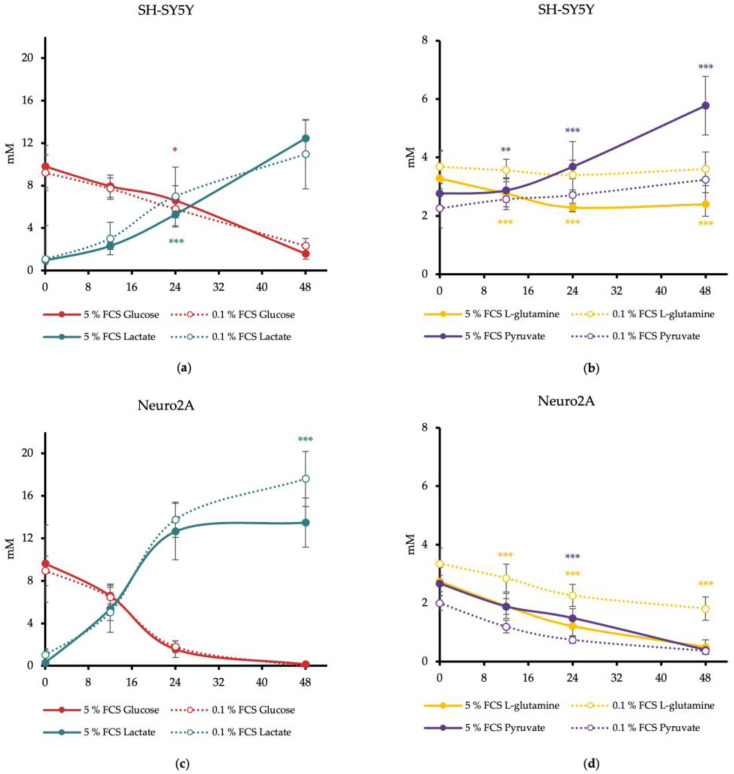
Concentrations of glucose, lactate, pyruvate, and L-glutamine in media of SH-SY5Y (**a**,**b**) and Neuro2A cells (**c**,**d**) in dependence of FCS concentration. Cells were treated with DMEM/5% FCS or DMEM/0.1% for up to 48 h. Concentrations of glucose, lactate, pyruvate and L-glutamine were measured at 0, 12, 24, and 48 h from media supernatant samples utilizing an Amplex Red Assay. *n* = 20. Substrate concentration in the cell culture medium is shown in mM. Error bars represent StDev. Statistical significance between 0.1% and 5% FCS is indicated by asterisks calculated by Student’s *t*-Test (* *p* ≤ 0.05, ** *p* ≤ 0.01, and *** *p* ≤ 0.001) (details see text).

**Figure 2 metabolites-14-00188-f002:**
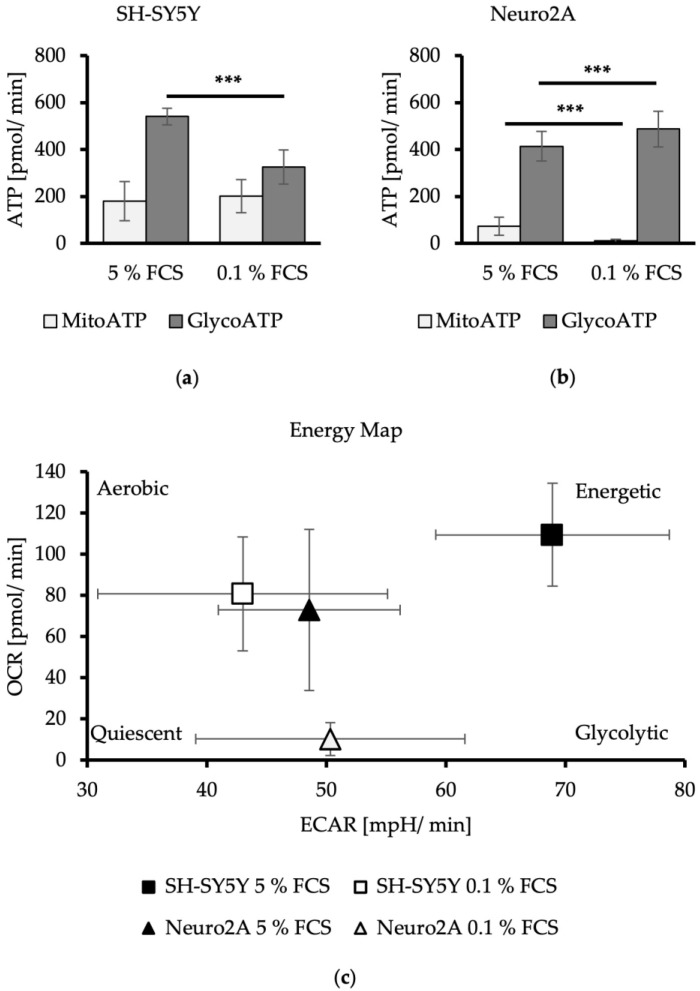
Aerobic and anaerobic ATP production level of SH-SY5Y (**a**) and Neuro2A cells (**b**) in dependence of FCS concentration in cell culture media. MitoATP = aerobic-produced ATP via mitochondrial oxidative phosphorylation in pmol/min. GlycoATP = anaerobic-produced ATP during glycolysis in pmol/min. (*n* = 24). (**c**) Energetic profiling of SH-SY5Y and Neuro2A. OCR (oxygen consumption rate) in pmol/min was applied against ECAR (extracellular acidification rate) in mpH/min (*n* = 24). Error bars represent StDev. Statistical significance between 0.1% and 5% FCS is indicated by asterisks calculated by Student´s *t*-Test (*** *p* ≤ 0.001) (details see text).

**Figure 3 metabolites-14-00188-f003:**
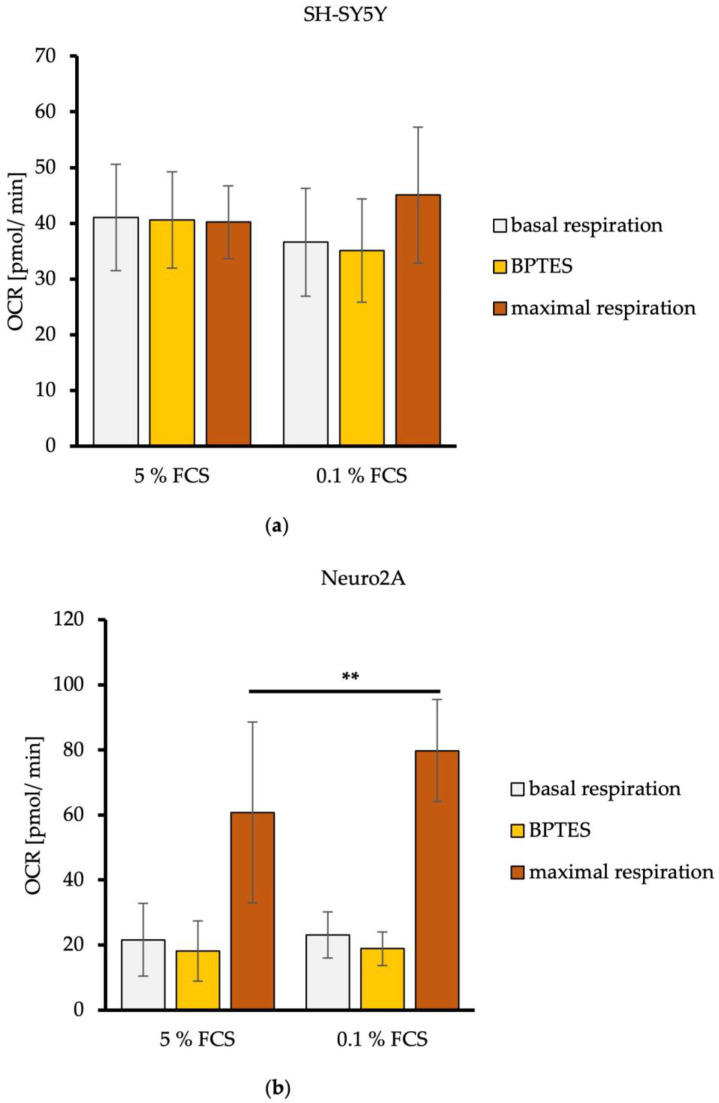
L-glutamine oxidation in dependence of FCS concentration by SH-SY5Y (**a**) and Neuro2A cells (**b**). Mitochondrial basal respiration under normal cellular substrate demand, acute response to L-glutamate deficiency triggered by Glutaminase 1 inhibitor BPTES (3 µM), and maximal respiration of L-glutamine under maximal mitochondrial substrate oxidation after injection of mitochondrial uncoupler FCCP (2 µM) of SH-SY5Y (**a**) and Neuro2A (**b**) cells treated with DMEM/5% FCS or DMEM/0.1% FCS (*n* = 24). OCR = oxygen consumption rate in pmol/min. Error bars represent StDev. Statistical significance between 0.1% and 5% FCS is indicated by asterisks calculated by Student’s *t*-Test (** *p* ≤ 0.01) (details see text).

**Table 1 metabolites-14-00188-t001:** Consumption rates (fmol/cell/h) of glucose, pyruvate, L-glutamine, and lactate. Values present consumption and production for the 0–24 h time window divided by 24.

	SH-SY5Y		Neuro2A	
[fmol/cell/h]	5% FCS	0.1% FCS	5% FCS	0.1% FCS
Glucose Consumption Rate	836.60	563.02	2093.72	2290.65
Pyruvate Consumption Rate	−239.75	−74.14	348.24	405.42
L-Glutamine Consumption Rate	152.00	46.69	239.85	180.18
Lactate Production Rate	1130.72	1349.63	3629.42	4137.20

## Data Availability

The datasets generated and analyzed during the current study are available from the corresponding author on reasonable request. The data are not publicly available due to privacy.

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
