# Peer review of "Metabolic Profiling of SH-SY5Y and Neuro2A Cells in Relation to Fetal Calf Serum (FCS) Concentration in Culture Media"

_metabolites, 2024, doi:10.3390/metabo14040188_

Round 1

Reviewer 1 Report

Comments and Suggestions for Authors The manuscript by Lys Kronenberger focuses on the changes in the basic ATP-generating pathways in neuroblastoma cell lines (SH-SY5Y and Neuro2A), with regard to the concentration of fetal calf serum. The authors demonstrate different reactions of cell metabolism after the decrease in FCS concentration. In the case of SH-SY5Y, stable mitochondrial ATP generation with a decrease in glycolytic ATP generation was shown. Neuro2A cells display a decrease in MitoATP and an increase in GlycoATP rates with stable total ATP generation rates. According to the authors, this compensation may be associated with a higher glutaminolysis level. It should be noted, however, that the manuscript requires some corrections: 1. The authors state in the text that "total ATP generation was not altered in this cell line", but this is not clearly evident from the data presented (Fig. 2B). It would be more accurate to report "total ATP generation rate" with the results of the statistics. 2. In Fig. 2c, energetic profiling is shown as a graph of OCR (oxygen consumption rate) against ECAR (energy consumption rate) in percentage. To help readers understand, it's best to explain what 100 percent means in terms of OCR and ECAR. 3. In Figs. 3a and 3b, the data is presented as OCR changes after glutaminase-1, while in Figs. 3c and 3d, the data for maximal OCR is shown. A unified approach would be recommended, as well as the data on basal respiration rates.

Author Response

Dear Reviewer 1,

Thank you very much for your time and effort to review our manuscript. Please find the detailed responses below and the corresponding corrections highlighted in the re-submitted files.

Comment 1:  The authors state in the text that "total ATP generation was not altered in this cell line", but this is not clearly evident from the data presented (Fig. 2B). It would be more accurate to report "total ATP generation rate" with the results of the statistics.

Response 1: Thank you for that advice. We fully agree with you. To visualize differences in the total ATP generation rate more clearly, we have inserted the mean values and standard deviation of the total ATP level for cells treated with DMEM/ 5 % FCS and DMEM/ 0.1 % FCS in lines 268-269 and 276-277 in the text after the respective statement on the effect of the FCS concentration on the total ATP level on pages 8 and 9.

Comment 2: In Fig. 2c, energetic profiling is shown as a graph of OCR (oxygen consumption rate) against ECAR (energy consumption rate) in percentage. To help readers understand, it's best to explain what 100 percent means in terms of OCR and ECAR.

Response 2: Thank you for giving the constructive comment. Therefore, we have changed the axes labeling of the OCR to pmol/ min and the ECAR to mpH/ min for a better understanding for the reader and a unified representation in Figure 2 on page 10.

Comment 3:  In Figs.3a and 3b, the data is presented as OCR changes after glutaminase-1, while in Figs. 3c and 3d, the data for maximal OCR is shown. A unified approach would be recommended, as well as the data on basal respiration rates.

Response 3: We thank you for your suggestion to expand and rearrange Figure 3. Therefore, we have included the basal respiration data. For a unified presentation of the data, we have shown the measured OCR in pmol/ min of basal respiration, respiration after injection of the inhibitor BPTES (acute response) and the maximal respiration for each of the two cell lines in one figure on pages 11-12.

Reviewer 2 Report

Comments and Suggestions for Authors

This manuscript, 'Metabolic Profiling of SH-SY5Y and Neuro2A Cells in Relation 2 to Fetal Calf Serum (FCS) Concentration in Culture Medium,' by Kronenberger and his group, shows the impact of nutrition on the metabolic profile of SH-SY5Y and 13 Neuro2A cells. The results of experiments indicate that changes in FCS concentration in culture medium affect the energetics and underlying biochemistry of SH-SY5Y and Neuro2A cells, which must be considered in future experimental studies. Overall, this is a well-done and clearly presented report. 

Minor comments

Line 91-92: should be 'passaged' and 'confluency'

Line 97: why was seeding density different in each condition? They should be kept constant to quantify L-glutamine in different media. 

Line 224, 228: The sentence is unclear and needs revision. 

 Line 267: divided instead.

Suggestions

Media affects the metabolic profile and the differentiation, growth, and molecular profile of SH-SY5Y cells. Authors might consider expanding their introduction and discussion by mentioning these areas as shown in doi: 10.3233/JAD-201409, doi: 10.1016/j.bbamcr.2020.118737.    

Author Response

Dear Reviewer 3,

Thank you very much for taking the time to review this manuscript. Please find the detailed responses below and the corresponding corrections highlighted in the re-submitted files.

Comment 1: Line 91-92: should be 'passaged' and 'confluency'

Response 1: Thank you for pointing this out. We have corrected the word spelling at this point.

Comment 2: Line 97: why was seeding density different in each condition? They should be kept constant to quantify L-glutamine in different media. 

Response 2: Thank you for giving this constructive comment. We have inserted an explanatory sentence why we have used different cell numbers in line 87-90.

Comment 3: Line 224, 228: The sentence is unclear and needs revision. 

Response 3: We thank you that you pointed this out. We have reviewed the sentences in line 217-230 and corrected the spelling for better understanding.

Comment 4: Line 267: divided instead.

Response 4: We thank you for this advice. We have expanded the description of Table 1 for a better understanding.

Comment 5: Media affects the metabolic profile and the differentiation, growth, and molecular profile of SH-SY5Y cells. Authors might consider expanding their introduction and discussion by mentioning these areas as shown in doi: 10.3233/JAD-201409, doi: 10.1016/j.bbamcr.2020.118737.

Response 5: Thank you for the interesting comment. Because another Reviewer has asked about the effects of retinoic acid, we have inserted two sentences in the discussion in line 369-372 to mention the possible effects of the differentiation of SH-SY5Y cells by retinoic acid (RA) on the metabolism of these cells and cited literature, which has investigated the effect of RA on mitochondrial oxidative consumption rate and glycolysis. Furthermore, we have pointed out the possible effects of treatments and media on cellular metabolism.

Reviewer 3 Report

Comments and Suggestions for Authors

Text seems quite basic about SH-SY5Y metabolism and well done to account with the basal functioning of the cells used for many tests and assays in toxicology and much more.

In lines 396-397 the term 'Plasmax' appeared without explanation. It was only after reading about that the reviewer had an idee of identity ... could you explain within the text what that meant?

Their differentiation with retinoic acid (RA) is frequent to involve them in cultivation tests. Have you other data about this RA effects on metabolism as it becomes more aerobic after RA? Or, could you comment because it is also a common differentiating agent (RA)?

Author Response

Dear Reviewer 2,

Thank you very much for taking the time to review this manuscript. Please find the detailed responses below and the corresponding corrections highlighted in the re-submitted files.

Comment 1: In lines 396-397 the term 'Plasmax' appeared without explanation. It was only after reading about that the reviewer had an idee of identity ... could you explain within the text what that meant?

Response 1: Thank you for pointing this out. We agree with this comment. Therefore, we have inserted a general explanatory sentence concerning the composition of Plasmax in lines 366-367 on page 12 after the first mention of the term Plasmax.

Comment 2: Their differentiation with retinoic acid (RA) is frequent to involve them in cultivation tests. Have you other data about this RA effects on metabolism as it becomes more aerobic after RA? Or, could you comment because it is also a common differentiating agent (RA)?

Response 2: Thank you for the interesting comment. Since there is already literature that has studied the effects of RA on the oxygen consumption rate in the mitochondria, for example (https://doi.org/10.1016/j.mad.2012.01.008), and we do not normally use RA in our laboratory because we only work with undifferentiated cells, we have focused on the effects of the FCS concentration in cell culture medium and not carried out any measurements with RA treatment and therefore have no data.
